# INCREMENTAL NETWORK QUANTIZATION: TOWARDS LOSSLESS CNNS WITH LOW-PRECISION WEIGHTS

**Aojun Zhou**[*]**, Anbang Yao, Yiwen Guo, Lin Xu, and Yurong Chen**
Intel Labs China
{aojun.zhou, anbang.yao, yiwen.guo, lin.x.xu, yurong.chen}@intel.com

## ABSTRACT

This paper presents incremental network quantization (INQ), a novel method, targeting to efficiently convert any pre-trained full-precision convolutional neural network (CNN) model into a low-precision version whose weights are constrained to be either powers of two or zero. Unlike existing methods which are struggled in noticeable accuracy loss, our INQ has the potential to resolve this issue, as benefiting from two innovations. On one hand, we introduce three interdependent operations, namely weight partition, group-wise quantization and re-training. A well-proven measure is employed to divide the weights in each layer of a pre-trained CNN model into two disjoint groups. The weights in the first group are responsible to form a low-precision base, thus they are quantized by a variable-length encoding method. The weights in the other group are responsible to compensate for the accuracy loss from the quantization, thus they are the ones to be re-trained. On the other hand, these three operations are repeated on the latest re-trained group in an iterative manner until all the weights are converted into low-precision ones, acting as an incremental network quantization and accuracy enhancement procedure. Extensive experiments on the ImageNet classification task using almost all known deep CNN architectures including AlexNet, VGG-16, GoogleNet and ResNets well testify the efficacy of the proposed method. Specifically, at 5-bit quantization (a variable-length encoding: 1 bit for representing zero value, and the remaining 4 bits represent at most 16 different values for the powers of two) [1], our models have improved accuracy than the 32-bit floating-point references. Taking ResNet-18 as an example, we further show that our quantized models with 4-bit, 3-bit and 2-bit ternary weights have improved or very similar accuracy against its 32-bit floating-point baseline. Besides, impressive results with the combination of network pruning and INQ are also reported. We believe that our method sheds new insights on how to make deep CNNs to be applicable on mobile or embedded devices. The code will be made publicly available.

## 1 INTRODUCTION

Deep convolutional neural networks (CNNs) have demonstrated record breaking results on a variety of computer vision tasks such as image classification (Krizhevsky et al., 2012; Simonyan & Zisserman, 2015), face recognition (Taigman et al., 2014; Sun et al., 2014), semantic segmentation (Long et al., 2015; Chen et al., 2015a) and object detection (Girshick, 2015; Ren et al., 2015). Regardless of the availability of significantly improved training resources such as abundant annotated data, powerful computational platforms and diverse training frameworks, the promising results of deep CNNs are mainly attributed to the large number of learnable parameters, ranging from tens of millions to even hundreds of millions. Recent progress further shows clear evidence that CNNs could easily enjoy the accuracy gain from the increased network depth and width (He et al., 2016; Szegedy et al., 2015; 2016). However, this in turn lays heavy burdens on the memory and other

---

[*]This work was done when Aojun Zhou was an intern at Intel Labs China, supervised by Anbang Yao who proposed the original idea and is responsible for correspondence. The first three authors contributed equally to the writing of the paper.

[1]This notation applies to our method throughout the paper.

computational resources. For instance, ResNet-152, a specific instance of the latest residual network architecture wining ImageNet classification challenge in 2015, has a model size of about 230 MB and needs to perform about 11.3 billion FLOPs to classify a $224 \times 224$ image crop. Therefore, it is very challenging to deploy deep CNNs on the devices with limited computation and power budgets.

Substantial efforts have been made to the speed-up and compression on CNNs during training, feed-forward test or both of them. Among existing methods, the category of network quantization methods attracts great attention from researches and developers. Some network quantization works try to compress pre-trained full-precision CNN models directly. Gong et al. (2014) address the storage problem of AlexNet (Krizhevsky et al., 2012) with vector quantization techniques. By replacing the weights in each of the three fully connected layers with respective floating-point centroid values obtained from the clustering, they can get over $20\times$ model compression at about 1% loss in top-5 recognition rate. HashedNet (Chen et al., 2015b) uses a hash function to randomly map pre-trained weights into hash buckets, and all the weights in the same hash bucket are constrained to share a single floating-point value. In HashedNet, only the fully connected layers of several shallow CNN models are considered. For better compression, Han et al. (2016) present deep compression method which combines the pruning (Han et al., 2015), vector quantization and Huffman coding, and reduce the model storage by $35\times$ on AlexNet and $49\times$ on VGG-16 (Simonyan & Zisserman, 2015). Vanhoucke et al. (2011) use an SSE 8-bit fixed-point implementation to improve the computation of neural networks on the modern Intel x86 CPUs in feed-forward test, yielding $3\times$ speed-up over an optimized floating-point baseline. Training CNNs by substituting the 32-bit floating-point representation with the 16-bit fixed-point representation has also been explored in Gupta et al. (2015). Other seminal works attempt to restrict CNNs into low-precision versions during training phase. Soudry et al. (2014) propose expectation backpropagation (EBP) to estimate the posterior distribution of deterministic network weights. With EBP, the network weights can be constrained to +1 and -1 during feed-forward test in a probabilistic way. BinaryConnect (Courbariaux et al., 2015) further extends the idea behind EBP to binarize network weights during training phase directly. It has two versions of network weights: floating-point and binary. The floating-point version is used as the reference for weight binarization. BinaryConnect achieves state-of-the-art accuracy using shallow CNNs for small datasets such as MNIST (LeCun et al., 1998) and CIFAR-10. Later on, a series of efforts have been invested to train CNNs with low-precision weights, low-precision activations and even low-precision gradients, including but not limited to BinaryNet (Courbariaux et al., 2016), XNOR-Net (Rastegari et al., 2016), ternary weight network (TWN) (Li & Liu, 2016), DoReFa-Net (Zhou et al., 2016) and quantized neural network (QNN) (Hubara et al., 2016).

Despite these tremendous advances, CNN quantization still remains an open problem due to two critical issues which have not been well resolved yet, especially under scenarios of using low-precision weights for quantization. The first issue is the non-negligible accuracy loss for CNN quantization methods, and the other issue is the increased number of training iterations for ensuring convergence. In this paper, we attempt to address these two issues by presenting a novel incremental network quantization (INQ) method.

In our INQ, there is no assumption on the CNN architecture, and its basic goal is to efficiently convert any pre-trained full-precision (i.e., 32-bit floating-point) CNN model into a low-precision version whose weights are constrained to be either powers of two or zero. The advantage of such kind of low-precision models is that the original floating-point multiplication operations can be replaced by cheaper binary bit shift operations on dedicated hardware like FPGA. We noticed that most existing network quantization methods adopt a global strategy in which all the weights are simultaneously converted to low-precision ones (that are usually in the floating-point types). That is, they have not considered the different importance of network weights, leaving the room to retain network accuracy limited. In sharp contrast to existing methods, our INQ makes a very careful handling for the model accuracy drop from network quantization. To be more specific, it incorporates three interdependent operations: weight partition, group-wise quantization and re-training. Weight partition uses a pruning-inspired measure (Han et al., 2015; Guo et al., 2016) to divide the weights in each layer of a pre-trained full-precision CNN model into two disjoint groups which play complementary roles in our INQ. The weights in the first group are quantized to be either powers of two or zero by a variable-length encoding method, forming a low-precision base for the original model. The weights in the other group are re-trained while keeping the quantized weights fixed, compensating for the accuracy loss resulted from the quantization. Furthermore, these three operations are repeated on the

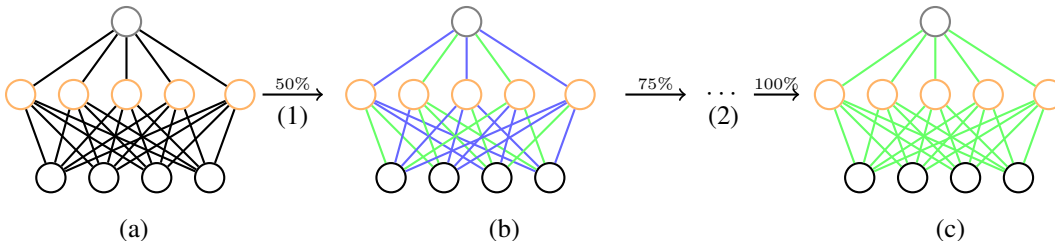

Figure 1: An overview of our incremental network quantization method. (a) Pre-trained full-precision model used as a reference. (b) Model update with three proposed operations: weight partition, group-wise quantization (green connections) and re-training (blue connections). (c) Final low-precision model with all the weights constrained to be either powers of two or zero. In the figure, operation (1) represents a single run of (b), and operation (2) denotes the procedure of repeating operation (1) on the latest re-trained weight group until all the non-zero weights are quantized. Our method does not lead to accuracy loss when using 5-bit, 4-bit and even 3-bit approximations in network quantization. For better visualization, here we just use a 3-layer fully connected network as an illustrative example, and the newly re-trained weights are divided into two disjoint groups of the same size at each run of operation (1) except the last run which only performs quantization on the re-trained floating-point weights occupying 12.5% of the model weights.

latest re-trained weight group in an iterative manner until all the weights are quantized, acting as an incremental network quantization and accuracy enhancement procedure (as illustrated in Figure 1).

The main insight of our INQ is that a compact combination of the proposed weight partition, group-wise quantization and re-training operations has the potential to get a lossless low-precision CNN model from any full-precision reference. We conduct extensive experiments on the ImageNet large scale classification task using almost all known deep CNN architectures to validate the effectiveness of our method. We show that: (1) For AlexNet, VGG-16, GoogleNet and ResNets with 5-bit quantization, INQ achieves improved accuracy in comparison with their respective full-precision baselines. The absolute top-1 accuracy gain ranges from **0.13%** to **2.28%**, and the absolute top-5 accuracy gain is in the range of **0.23%** to **1.65%**. (2) INQ has the property of easy convergence in training. In general, re-training with less than **8 epochs** could consistently generate a lossless model with 5-bit weights in the experiments. (3) Taking ResNet-18 as an example, our quantized models with 4-bit, 3-bit and 2-bit ternary weights also have improved or very similar accuracy compared with its 32-bit floating-point baseline. (4) Taking AlexNet as an example, the combination of our network pruning and INQ outperforms deep compression method (Han et al., 2016) with significant margins.

## 2 INCREMENTAL NETWORK QUANTIZATION

In this section, we clarify the insight of our INQ, describe its key components, and detail its implementation.

### 2.1 WEIGHT QUANTIZATION WITH VARIABLE-LENGTH ENCODING

Suppose a pre-trained full-precision (i.e., 32-bit floating-point) CNN model can be represented by $\{\mathbf{W}_l : 1 \leq l \leq L\}$, where $\mathbf{W}_l$ denotes the weight set of the $l^{th}$ layer, and $L$ denotes the number of learnable layers in the model. To simplify the explanation, we only consider convolutional layers and fully connected layers. For CNN models like AlexNet, VGG-16, GoogleNet and ResNets as tested in this paper, $\mathbf{W}_l$ can be a $4D$ tensor for the convolutional layer, or a $2D$ matrix for the fully connected layer. For simplicity, here the dimension difference is not considered in the expression. Given a pre-trained full-precision CNN model, the main goal of our INQ is to convert all 32-bit floating-point weights to be either powers of two or zero without loss of model accuracy. Besides, we also attempt to explore the limit of the expected bit-width under the premise of guaranteeing lossless network quantization. Here, we start with our basic network quantization method on how to

convert $\mathbf{W}_l$ to be a low-precision version $\widehat{\mathbf{W}}_l$, and each of its entries is chosen from

$$\mathbf{P}_l = \{\pm 2^{n_1}, \cdots, \pm 2^{n_2}, 0\}, \tag{1}$$

where $n_1$ and $n_2$ are two integer numbers, and they satisfy $n_2 \leq n_1$. Mathematically, $n_1$ and $n_2$ help to bound $\mathbf{P}_l$ in the sense that its non-zero elements are constrained to be in the range of either $[-2^{n_1}, -2^{n_2}]$ or $[2^{n_2}, 2^{n_1}]$. That is, network weights with absolute values smaller than $2^{n_2}$ will be pruned away (i.e., set to zero) in the final low-precision model. Obviously, the problem is how to determine $n_1$ and $n_2$. In our INQ, the expected bit-width $b$ for storing the indices in $\mathbf{P}_l$ is set beforehand, thus the only hyper-parameter shall be determined is $n_1$ because $n_2$ can be naturally computed once $b$ and $n_1$ are available. Here, $n_1$ is calculated by using a tricky yet practically effective formula as

$$n_1 = \mathrm{floor}(\log_2(4s/3)), \tag{2}$$

where $\mathrm{floor}(\cdot)$ indicates the round down operation and $s$ is calculated by using

$$s = \max(\mathrm{abs}(\mathbf{W}_l)), \tag{3}$$

where $\mathrm{abs}(\cdot)$ is an element-wise operation and $\max(\cdot)$ outputs the largest element of its input. In fact, Equation (2) helps to match the rounding power of 2 for $s$, and it could be easily implemented in practical programming. After $n_1$ is obtained, $n_2$ can be naturally determined as $n_2 = n_1 + 1 - 2^{(b-1)}/2$. For instance, if $b = 3$ and $n_1 = -1$, it is easy to get $n_2 = -2$.

Once $\mathbf{P}_l$ is determined, we further use the ladder of powers to convert every entry of $\mathbf{W}_l$ into a low-precision one by using

$$\widehat{\mathbf{W}}_l(i,j) = \begin{cases} \beta \mathrm{sgn}(\mathbf{W}_l(i,j)) & \text{if } (\alpha + \beta)/2 \leq \mathrm{abs}(\mathbf{W}_l(i,j)) < 3\beta/2 \\ 0 & \text{otherwise,} \end{cases} \tag{4}$$

where $\alpha$ and $\beta$ are two adjacent elements in the sorted $\mathbf{P}_l$, making the above equation as a numerical rounding to the quantum values. It should be emphasized that factor $4/3$ in Equation (2) is set to make sure that all the elements in $\mathbf{P}_l$ correspond with the quantization rule defined in Equation (4). In other words, factor $4/3$ in Equation (2) highly correlates with factor $3/2$ in Equation (4).

Here, an important thing we want to clarify is the definition of the expected bit-width $b$. Taking 5-bit quantization as an example, since zero value cannot be written as the power of two, we use 1 bit to represent zero value, and the remaining 4 bits to represent at most 16 different values for the powers of two. That is, the number of candidate quantum values is at most $2^{b-1} + 1$, so our quantization method actually adopts a variable-length encoding scheme. It is clear that the quantization described above is performed in a linear scale. An alternative solution is to perform the quantization in the log scale. Although it may also be effective, it should be a little bit more difficult in implementation and may cause some extra computational overhead in comparison to our method.

## 2.2 INCREMENTAL QUANTIZATION STRATEGY

We can naturally use the above described method to quantize any pre-trained full-precision CNN model. However, noticeable accuracy loss appeared in the experiments when using small bit-width values (e.g., 5-bit, 4-bit, 3-bit and 2-bit).

In the literature, there are many existing network quantization works such as HashedNet (Chen et al., 2015b), vector quantization (Gong et al., 2014), fixed-point representation (Vanhoucke et al., 2011; Gupta et al., 2015), BinaryConnect (Courbariaux et al., 2015), BinaryNet (Courbariaux et al., 2016), XNOR-Net (Rastegari et al., 2016), TWN (Li & Liu, 2016), DoReFa-Net (Zhou et al., 2016) and QNN (Hubara et al., 2016). Similar to our basic network quantization method, they also suffer from non-negligible accuracy loss on deep CNNs, especially when being applied on the ImageNet large scale classification dataset. For all these methods, a common fact is that they adopt a global strategy in which all the weights are simultaneously converted into low-precision ones, which in turn causes accuracy loss. Compared with the methods focusing on the pre-trained models, accuracy loss becomes worse for the methods such as XNOR-Net, TWN, DoReFa-Net and QNN which intend to train low-precision CNNs from scratch.

Recall that our main goal is to achieve lossless low-precision quantization for any pre-trained full-precision CNN model with no assumption on its architecture. To this end, our INQ makes a special

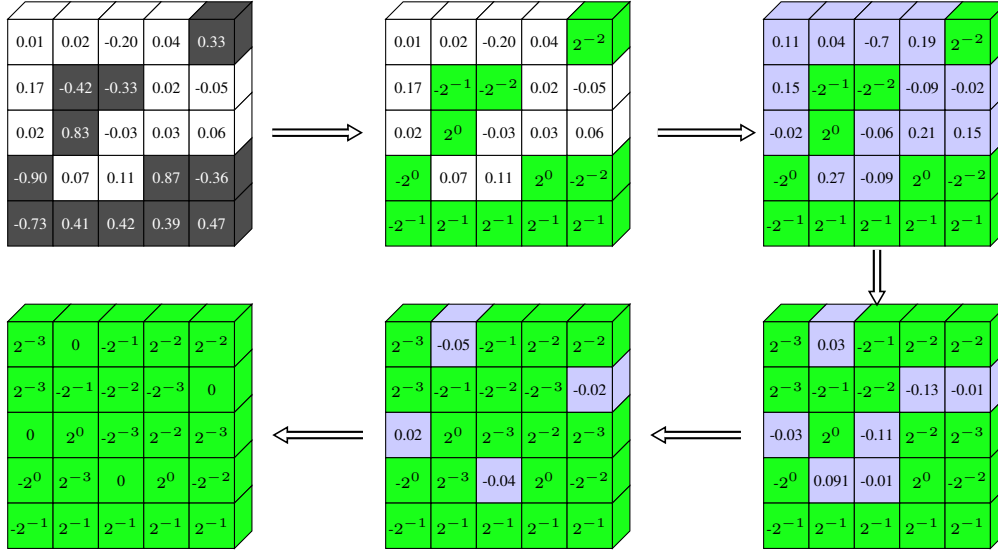

Figure 2: Result illustrations. First row: results from the $1^{st}$ iteration of the proposed three operations. The top left cube illustrates weight partition operation generating two disjoint groups, the middle image illustrates the quantization operation on the first weight group (green cells), and the top right cube illustrates the re-training operation on the second weight group (light blue cells). Second row: results from the $2^{nd}$, $3^{rd}$ and $4^{th}$ iterations of the INQ. In the figure, the accumulated portion of the weights which have been quantized undergoes from 50%→75%→87.5%→100%.

handling of the strategy for suppressing resulting quantization loss in model accuracy. We are partially inspired by the latest progress in network pruning (Han et al., 2015; Guo et al., 2016). In these methods, the accuracy loss from removing less important network weights of a pre-trained neural network model could be well compensated by following re-training steps. Therefore, we conjecture that the nature of changing network weight importance is critical to achieve lossless network quantization.

Base on this assumption, we present INQ which incorporates three interdependent operations: weight partition, group-wise quantization and re-training. Weight partition is to divide the weights in each layer of a pre-trained full-precision CNN model into two disjoint groups which play complementary roles in our INQ. The weights in the first group are responsible for forming a low-precision base for the original model, thus they are quantized by using Equation (4). The weights in the second group adapt to compensate for the loss in model accuracy, thus they are the ones to be re-trained. Once the first run of the quantization and re-training operations is finished, all the three operations are further conducted on the second weight group in an iterative manner, until all the weights are converted to be either powers of two or zero, acting as an incremental network quantization and accuracy enhancement procedure. As a result, accuracy loss under low-precision CNN quantization can be well suppressed by our INQ. Illustrative results at iterative steps of our INQ are provided in Figure 2.

For the $l^{th}$ layer, weight partition can be defined as

$$\mathbf{A}_l^{(1)} \cup \mathbf{A}_l^{(2)} = \{\mathbf{W}_l(i,j)\}, \text{ and } \mathbf{A}_l^{(1)} \cap \mathbf{A}_l^{(2)} = \emptyset, \tag{5}$$

where $\mathbf{A}_l^{(1)}$ denotes the first weight group that needs to be quantized, and $\mathbf{A}_2$ denotes the other weight group that needs to be re-trained. **We leave the strategies for group partition to be chosen in the experiment section**. Here, we define a binary matrix $\mathbf{T}_l$ to help distinguish above two categories of weights. That is, $\mathbf{T}_l(i,j) = 0$ means $\mathbf{W}_l(i,j) \in \mathbf{A}_l^{(1)}$, and $\mathbf{T}_l(i,j) = 1$ means $\mathbf{W}_l(i,j) \in \mathbf{A}_l^{(2)}$.

## 2.3 INCREMENTAL NETWORK QUANTIZATION ALGORITHM

Now, we come to the training method. Taking the $l^{th}$ layer as an example, the basic optimization problem of making its weights to be either powers of two or zero can be expressed as

$$
\begin{aligned}
\min_{\mathbf{W}_l} \quad & E(\mathbf{W}_l) = L(\mathbf{W}_l) + \lambda R(\mathbf{W}_l) \\
\text{s.t.} \quad & \mathbf{W}_l(i,j) \in \mathbf{P}_l, \ 1 \le l \le L,
\end{aligned}
\tag{6}
$$

where $L(\mathbf{W}_l)$ is the network loss, $R(\mathbf{W}_l)$ is the regularization term, $\lambda$ is a positive coefficient, and the constraint term indicates each weight entry $\mathbf{W}_l(i,j)$ should be chosen from the set $\mathbf{P}_l$ consisting of a fixed number of the values of powers of two plus zero. Direct solving above optimization problem in training from scratch is challenging since it is very easy to undergo convergence problem.

By performing weight partition and group-wise quantization operations beforehand, the optimization problem defined in (6) can be reshaped into a easier version. That is, we only need to optimize the following objective function

$$
\begin{aligned}
\min_{\mathbf{W}_l} \quad & E(\mathbf{W}_l) = L(\mathbf{W}_l) + \lambda R(\mathbf{W}_l) \\
\text{s.t.} \quad & \mathbf{W}_l(i,j) \in \mathbf{P}_l, \ \text{if } \mathbf{T}_l(i,j) = 0, \ 1 \le l \le L,
\end{aligned}
\tag{7}
$$

where $\mathbf{P}_l$ is determined at group-wise quantization operation, and the binary matrix $\mathbf{T}_l$ acts as a mask which is determined by weight partition operation. Since $\mathbf{P}_l$ and $\mathbf{T}_l$ are known, the optimization problem (7) can be solved using popular stochastic gradient decent (SGD) method. That is, in INQ, we can get the update scheme for the re-training as

$$
\mathbf{W}_l(i,j) \leftarrow \mathbf{W}_l(i,j) - \gamma \frac{\partial E}{\partial(\mathbf{W}_l(i,j))} \mathbf{T}_l(i,j),
\tag{8}
$$

where $\gamma$ is a positive learning rate. Note that the binary matrix $\mathbf{T}_l$ forces zero update to the weights that have been quantized. That is, only the weights still keep with floating-point values are updated, akin to the latest pruning methods (Han et al., 2015; Guo et al., 2016) in which only the weights that are not currently removed are re-trained to enhance network accuracy. The whole procedure of our INQ is summarized as Algorithm 1.

We would like to highlight that the merits of our INQ are in three aspects: (1) Weight partition introduces the importance-aware weight quantization. (2) Group-wise weight quantization introduces much less accuracy loss than simultaneously quantizing all the network weights, thus making re-training have larger room to recover model accuracy. (3) By integrating the operations of weight partition, group-wise quantization and re-training into a nested loop, our INQ has the potential to obtain lossless low-precision CNN model from the pre-trained full-precision reference.

---

**Algorithm 1** Incremental network quantization for lossless CNNs with low-precision weights.

---

**Input:** $X$: the training data, $\{\mathbf{W}_l : 1 \le l \le L\}$: the pre-trained full-precision CNN model, $\{\sigma_1, \sigma_2, \cdots, \sigma_N\}$: the accumulated portions of weights quantized at iterative steps

**Output:** $\{\widehat{\mathbf{W}}_l : 1 \le l \le L\}$: the final low-precision model with the weights constrained to be either powers of two or zero

1: Initialize $\mathbf{A}_l^{(1)} \leftarrow \emptyset$, $\mathbf{A}_l^{(2)} \leftarrow \{\mathbf{W}_l(i,j)\}$, $\mathbf{T}_l \leftarrow \mathbf{1}$, for $1 \le l \le L$
2: **for** $n = 1, 2, \ldots, N$ **do**
3: Reset the base learning rate and the learning policy
4: According to $\sigma_n$, perform layer-wise weight partition and update $\mathbf{A}_l^{(1)}$, $\mathbf{A}_l^{(2)}$ and $\mathbf{T}_l$
5: Based on $\mathbf{A}_l^{(1)}$, determine $\mathbf{P}_l$ layer-wisely
6: Quantize the weights in $\mathbf{A}_l^{(1)}$ by Equation (4) layer-wisely
7: Calculate feed-forward loss, and update weights in $\{\mathbf{A}_l^{(2)} : 1 \le l \le L\}$ by Equation (8)
8: **end for**

---

## 3 EXPERIMENTAL RESULTS

To analyze the performance of our INQ, we perform extensive experiments on the ImageNet large scale classification task, which is known as the most challenging image classification benchmark so far. ImageNet dataset has about 1.2 million training images and 50 thousand validation images. Each image is annotated as one of 1000 object classes. We apply our INQ to AlexNet, VGG-16, GoogleNet, ResNet-18 and ResNet-50, covering almost all known deep CNN architectures. Using the center crops of validation images, we report the results with two standard measures: top-1 error rate and top-5 error rate. For fair comparison, all pre-trained full-precision (i.e., 32-bit floating-point) CNN models except ResNet-18 are taken from the Caffe model zoo[2]. Note that He et al. (2016) do not release their pre-trained ResNet-18 model to the public, so we use a publicly available re-implementation by Facebook[3]. Since our method is implemented with Caffe, we make use of an open source tool[4] to convert the pre-trained ResNet-18 model from Torch to Caffe.

### 3.1 RESULTS ON IMAGENET

Table 1: Our INQ well converts diverse full-precision deep CNN models (including AlexNet, VGG-16, GoogleNet, ResNet-18 and ResNet-50) to 5-bit low-precision versions with consistently improved model accuracy.

| Network | Bit-width | Top-1 error | Top-5 error | Decrease in top-1/top-5 error |
|---|---|---|---|---|
| AlexNet ref | 32 | 42.76% | 19.77% | |
| AlexNet | 5 | **42.61%** | **19.54%** | 0.15%/0.23% |
| VGG-16 ref | 32 | 31.46% | 11.35% | |
| VGG-16 | 5 | **29.18%** | **9.70%** | 2.28%/1.65% |
| GoogleNet ref | 32 | 31.11% | 10.97% | |
| GoogleNet | 5 | **30.98%** | **10.72%** | 0.13%/0.25% |
| ResNet-18 ref | 32 | 31.73% | 11.31% | |
| ResNet-18 | 5 | **31.02%** | **10.90%** | 0.71%/0.41% |
| ResNet-50 ref | 32 | 26.78% | 8.76% | |
| ResNet-50 | 5 | **25.19%** | **7.55%** | 1.59%/1.21% |

Setting expected bit-width to 5, the first set of experiments is performed to testify the efficacy of our INQ on different CNN architectures. Regarding weight partition, there are several candidate strategies as we tried in our previous work for efficient network pruning (Guo et al., 2016). In Guo et al. (2016), we found random partition and pruning-inspired partition are the two best choices compared with the others. Thus in this paper, we directly compare these two strategies for weight partition. In random strategy, the weights in each layer of any pre-trained full-precision deep CNN model are randomly split into two disjoint groups. In pruning-inspired strategy, the weights are divided into two disjoint groups by comparing their absolute values with layer-wise thresholds which are automatically determined by a given splitting ratio. Here we directly use pruning-inspired strategy and the experimental results in Section 3.2 will show why. After the re-training with no more than 8 epochs over each pre-trained full-precision model, we obtain the results as shown in Table 1. It can be concluded that the 5-bit CNN models generated by our INQ show consistently improved top-1 and top-5 recognition rates compared with respective full-precision references. Parameter settings are described below.

**AlexNet:** AlexNet has 5 convolutional layers and 3 fully-connected layers. We set the accumulated portions of quantized weights at iterative steps as {0.3, 0.6, 0.8, 1}, the batch size as 256, the weight decay as 0.0005, and the momentum as 0.9.

**VGG-16:** Compared with AlexNet, VGG-16 has 13 convolutional layers and more parameters. We set the accumulated portions of quantized weights at iterative steps as {0.5, 0.75, 0.875, 1}, the batch size as 32, the weight decay as 0.0005, and the momentum as 0.9.

---

[2]https://github.com/BVLC/caffe/wiki/Model-Zoo

[3]https://github.com/facebook/fb.resnet.torch/tree/master/pretrained

[4]https://github.com/zhanghang1989/fb-caffe-exts

**GoogleNet:** Compared with AlexNet and VGG-16, GoogleNet is more difficult to quantize due to a smaller number of parameters and the increased network width. We set the accumulated portions of quantized weights at iterative steps as {0.2, 0.4, 0.6, 0.8, 1}, the batch size as 80, the weight decay as 0.0002, and the momentum as 0.9.

**ResNet-18:** Different from above three networks, ResNets have batch normalization layers and relief the vanishing gradient problem by using shortcut connections. We first test the 18-layer version for exploratory purpose and test the 50-layer version later on. The network architectures of ResNet-18 and ResNet-34 are very similar. The only difference is the number of filters in every convolutional layer. We set the accumulated portions of quantized weights at iterative steps as {0.5, 0.75, 0.875, 1}, the batch size as 80, the weight decay as 0.0005, and the momentum as 0.9.

**ResNet-50:** Besides significantly increased network depth, ResNet-50 has a more complex network architecture in comparison to ResNet-18. However, regarding network architecture, ResNet-50 is very similar to ResNet-101 and ResNet-152. The only difference is the number of filters in every convolutional layer. We set the accumulated portions of quantized weights at iterative steps as {0.5, 0.75, 0.875, 1}, the batch size as 32, the weight decay as 0.0005, and the momentum as 0.9.

## 3.2 ANALYSIS OF WEIGHT PARTITION STRATEGIES

In our INQ, the first operation is weight partition whose result will directly affect the following group-wise quantization and re-training operations. Therefore, the second set of experiments is conducted to analyze two candidate strategies for weight partition. As mentioned in the previous section, we use pruning-inspired strategy for weight partition. Unlike random strategy in which all the weights have equal probability to fall into the two disjoint groups, pruning-inspired strategy considers that the weights with larger absolute values are more important than the smaller ones to form a low-precision base for the original CNN model. We use ResNet-18 as a test case to compare the performance of these two strategies. In the experiments, the parameter settings are completely the same as described in Section 3.1. We set 4 epochs for weight re-training. Table 2 summarizes the results of our INQ with 5-bit quantization. It can be seen that our INQ achieves top-1 error rate of 32.11% and top-5 error rate of 11.73% by using random partition. Comparatively, pruning-inspired partition brings 1.09% and 0.83% decrease in top-1 and top-5 error rates, respectively. Apparently, pruning-inspired partition is better than random partition, and this is the reason why we use it in this paper. For future works, weight partition based on quantization error could also be an option worth exploring.

Table 2: Comparison of two different strategies for weight partition on ResNet-18.

| Strategy | Bit-width | Top-1 error | Top-5 error |
|---|---|---|---|
| Random partition | 5 | 32.11% | 11.73% |
| Pruning-inspired partition | 5 | **31.02%** | **10.90%** |

## 3.3 THE TRADE-OFF BETWEEN EXPECTED BIT-WIDTH AND MODEL ACCURACY

The third set of experiments is performed to explore the limit of the expected bit-width under which our INQ can still achieve lossless network quantization. Similar to the second set of experiments, we also use ResNet-18 as a test case, and the parameter settings for the batch size, the weight decay and the momentum are completely the same. Finally, lower-precision models with 4-bit, 3-bit and even 2-bit ternary weights are generated for comparisons. As the expected bit-width goes down, the number of candidate quantum values will be decreased significantly, thus we shall increase the number of iterative steps accordingly for enhancing the accuracy of final low-precision model. Specifically, we set the accumulated portions of quantized weights at iterative steps as {0.3, 0.5, 0.8, 0.9, 0.95, 1}, {0.2, 0.4, 0.6, 0.7, 0.8, 0.9, 0.95, 1} and {0.2, 0.4, 0.6, 0.7, 0.8, 0.85, 0.9, 0.95, 0.975, 1} for 4-bit, 3-bit and 2-bit ternary models, respectively. The required number of epochs also increases when the expected bit-width goes down, and it reaches 30 when training our 2-bit ternary model. Although our 4-bit model shows slightly decreased accuracy when compared with the 5-bit model, its accuracy is still better than that of the pre-trained full-precision model. Comparatively, even when the expected bit-width goes down to 3, our low-precision model shows only 0.19% and

0.33% losses in top-1 and top-5 recognition rates, respectively. As for our 2-bit ternary model, although it incurs 2.25% decrease in top-1 error rate and 1.56% decrease in top-5 error rate in comparison to the pre-trained full-precision reference, its accuracy is considerably better than state-of-the-art results reported for binary-weight network (BWN) (Rastegari et al., 2016) and ternary weight network (TWN) (Li & Liu, 2016). Detailed results are summarized in Table 3 and Table 4.

Table 3: Our INQ generates extremely low-precision (4-bit and 3-bit) models with improved or very similar accuracy compared with the full-precision ResNet-18 model.

| Model | Bit-width | Top-1 error | Top-5 error |
|---|---|---|---|
| ResNet-18 ref | 32 | 31.73% | 11.31% |
| INQ | 5 | 31.02% | 10.90% |
| INQ | 4 | 31.11% | 10.99% |
| INQ | 3 | 31.92% | 11.64% |
| INQ | 2 (ternary) | 33.98% | 12.87% |

Table 4: Comparison of our 2-bit ternary model and some other binary or ternary models, including the BWN and the TWN approximations of ResNet-18.

| Method | Bit-width | Top-1 error | Top-5 error |
|---|---|---|---|
| BWN(Rastegari et al., 2016) | 1 | 39.20% | 17.00% |
| TWN(Li & Liu, 2016) | 2 (ternary) | 38.20% | 15.80% |
| INQ (ours) | 2 (ternary) | **33.98%** | **12.87%** |

## 3.4 Low-Bit Deep Compression

In the literature, recently proposed deep compression method (Han et al., 2016) reports so far best results on network compression without loss of model accuracy. Therefore, the last set of experiments is conducted to explore the potential of our INQ for much better deep compression. Note that Han et al. (2016) is a hybrid network compression solution combining three different techniques, namely network pruning (Han et al., 2015), vector quantization (Gong et al., 2014) and Huffman coding. Taking AlexNet as an example, network pruning gets 9× compression, however this result is mainly obtained from the fully connected layers. Actually its compression performance on the convolutional layers is less than 3× (as can be seen in the Table 4 of Han et al. (2016)). Besides, network pruning is realized by separately performing pruning and re-training in an iterative way, which is very time-consuming. It will cost at least several weeks for compressing AlexNet. We solved this problem by our dynamic network surgery (DNS) method (Guo et al., 2016) which achieves about 7× speed-up in training and improves the performance of network pruning from 9× to 17.7×. In Han et al. (2016), after network pruning, vector quantization further improves compression ratio from 9× to 27×, and Huffman coding finally boosts compression ratio up to 35×. For fair comparison, we combine our proposed INQ and DNS, and compare the resulting method with Han et al. (2016). Detailed results are summarized in Table 5. When combing our proposed INQ and DNS, we achieve much better compression results compared with Han et al. (2016). Specifically, with 5-bit quantization, we can achieve 53× compression with slightly larger gains both in top-5 and top-1 recognition rates, yielding 51.43%/96.30% absolute improvement in compression performance compared with full version/fair version (i.e., the combination of network pruning and vector quantization) of Han et al. (2016), respectively. Consistently better results have also obtained for our 4-bit and 3-bit models.

Besides, we also perform a set of experiments on AlexNet to compare the performance of our INQ and vector quantization (Gong et al., 2014). For fair comparison, re-training is also used to enhance the performance of vector quantization, and we set the number of cluster centers for all of 5 convolutional layers and 3 fully connect layers to 32 (i.e., 5-bit quantization). In the experiment, vector quantization incurs over 3% loss in model accuracy. When we change the number of cluster centers for convolutional layers from 32 to 128, it gets an accuracy loss of 0.98%. This is consistent with the results reported in (Gong et al., 2014). Comparatively, vector quantization is mainly proposed

Table 5: Comparison of the combination of our INQ and DNS, and deep compression method on AlexNet. Conv: Convolutional layer, FC: Fully connected layer, P: Pruning, Q: Quantization, H: Huffman coding.

| Method | Bit-width(Conv/FC) | Compression ratio | Decrease in top-1/top5 error |
|---|---|---|---|
| Han et al. (2016) (P+Q) | 8/5 | 27× | 0.00%/0.03% |
| Han et al. (2016) (P+Q+H) | 8/5 | 35× | 0.00%/0.03% |
| Han et al. (2016) (P+Q+H) | 8/4 | - | -0.01%/0.00% |
| Our method (P+Q) | 5/5 | **53×** | **0.08%/0.03%** |
| Han et al. (2016) (P+Q+H) | 4/2 | - | -1.99%/-2.60% |
| Our method (P+Q) | 4/4 | **71×** | **-0.52%/-0.20%** |
| Our method (P+Q) | 3/3 | **89×** | **-1.47%/-0.96%** |

to compress the parameters in the fully connected layers of a pre-trained full-precision CNN model, while our INQ addresses all network layers simultaneously and has no accuracy loss for 5-bit and 4-bit quantization. Therefore, it is evident that our INQ is much better than vector quantization. Last but not least, the final weights for vector quantization (Gong et al., 2014), network pruning (Han et al., 2015) and deep compression (Han et al., 2016) are still floating-point values, but the final weights for our INQ are in the form of either powers of two or zero. The direct advantage of our INQ is that the original floating-point multiplication operations can be replaced by cheaper binary bit shift operations on dedicated hardware like FPGA.

## 4 CONCLUSIONS

In this paper, we present INQ, a new network quantization method, to address the problem of how to convert any pre-trained full-precision (i.e., 32-bit floating-point) CNN model into a lossless low-precision version whose weights are constrained to be either powers of two or zero. Unlike existing methods which usually quantize all the network weights simultaneously, INQ is a more compact quantization framework. It incorporates three interdependent operations: weight partition, group-wise quantization and re-training. Weight partition splits the weights in each layer of a pre-trained full-precision CNN model into two disjoint groups which play complementary roles in INQ. The weights in the first group is directly quantized by a variable-length encoding method, forming a low-precision base for the original CNN model. The weights in the other group are re-trained while keeping all the quantized weights fixed, compensating for the accuracy loss from network quantization. More importantly, the operations of weight partition, group-wise quantization and re-training are repeated on the latest re-trained weight group in an iterative manner until all the weights are quantized, acting as an incremental network quantization and accuracy enhancement procedure. On the ImageNet large scale classification task, we conduct extensive experiments and show that our quantized CNN models with 5-bit, 4-bit, 3-bit and even 2-bit ternary weights have improved or at least comparable accuracy against their full-precision baselines, including AlexNet, VGG-16, GoogleNet and ResNets. As for future works, we plan to extend incremental idea behind INQ from low-precision weights to low-precision activations and low-precision gradients (we have actually already made some good progress on it, as shown in our supplementary materials). We will also investigate computation and power efficiency by implementing our low-precision CNN models on hardware platforms.

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

## A    APPENDIX 1: STATISTICAL ANALYSIS OF THE QUANTIZED WEIGHTS

Taking our 5-bit AlexNet model as an example, we analyze the distribution of the quantized weights. Detailed statistical results are summarized in Table 6. We can find: (1) in the $1^{st}$ and $2^{nd}$ convolutional layers, the values of $\{-2^{-6}, -2^{-5}, -2^{-4}, 2^{-6}, 2^{-5}, 2^{-4}\}$ and $\{-2^{-8}, -2^{-7}, -2^{-6}, -2^{-5}, 0, 2^{-8}, 2^{-7}, 2^{-6}, 2^{-5}\}$ occupy over 60% and 94% of all quantized weights, respectively; (2) the distributions of the quantized weights in the $3^{rd}$, $4^{th}$ and $5^{th}$ convolutional layers are similar to that of the $2^{nd}$ convolutional layer, and more weights are quantized into zero in the $2^{nd}$, $3^{rd}$, $4^{th}$ and $5^{th}$ convolutional layers compared with the $1^{st}$ convolutional layer; (3) in the $1^{st}$ fully connected layer, the values of $\{-2^{-10}, -2^{-9}, -2^{-8}, -2^{-7}, 0, 2^{-10}, 2^{-9}, 2^{-8}, 2^{-7}\}$ occupy about 98% of all quantized weights, and similar results can be seen for the $2^{nd}$ fully connected layer; (4) generally, the distributions of the quantized weights in the convolutional layers are usually more scattered compared with the fully connected layers. This may be partially the reason why it is much easier to get good compression performance on fully connected layers in comparison to convolutional layers, when using methods such as network hashing (Chen et al., 2015b) and vector quantization (Gong et al., 2014); (5) for 5-bit AlexNet model, the required bit-width for each layer is actually 4 but not 5.

Table 6: A statistical distribution of the quantized weights in our 5-bit AlexNet model.

| Weight | Conv1 | Conv2 | Conv3 | Conv4 | Conv5 | FC6 | FC7 | FC8 |
|---|---|---|---|---|---|---|---|---|
| $-2^{-10}$ | - | - | - | - | - | 8.95% | 6.37% | 3.86% |
| $-2^{-9}$ | - | - | - | - | - | 12.29% | 9.58% | 6.19% |
| $-2^{-8}$ | 5.04% | 10.55% | 11.58% | 10.09% | 9.88% | 16.48% | 16.13% | 12.90% |
| $-2^{-7}$ | 6.56% | 12.09% | 14.34% | 14.24% | 14.68% | 10.84% | 17.87% | 19.51% |
| $-2^{-6}$ | 9.22% | 13.08% | 15.26% | 18.49% | 20.66% | 0.79% | 3.43% | 11.09% |
| $-2^{-5}$ | 10.52% | 8.73% | 5.92% | 7.77% | 9.79% | 0.002% | 0.004% | 0.40% |
| $-2^{-4}$ | 9.75% | 2.70% | 0.49% | 0.38% | 0.55% | - | - | - |
| $-2^{-3}$ | 4.61% | 0.39% | 0.02% | 0.01% | 0.004% | - | - | - |
| $-2^{-2}$ | 0.67% | 0.01% | $1e^{-4}\%$ | - | - | - | - | - |
| $0$ | 5.51% | 11.30% | 12.24% | 9.70% | 8.97% | 8.86% | 6.17% | 3.62% |
| $2^{-10}$ | - | - | - | - | - | 8.30% | 5.81% | 3.40% |
| $2^{-9}$ | - | - | - | - | - | 10.51% | 7.84% | 4.69% |
| $2^{-8}$ | 5.20% | 9.70% | 10.44% | 8.60% | 7.69% | 12.91% | 11.30% | 8.08% |
| $2^{-7}$ | 6.79% | 11.01% | 11.66% | 10.33% | 8.95% | 8.95% | 11.90% | 10.94% |
| $2^{-6}$ | 9.99% | 11.05% | 11.86% | 12.25% | 10.67% | 1.12% | 3.54% | 12.56% |
| $2^{-5}$ | 11.15% | 6.57% | 5.22% | 6.81% | 6.37% | 0.01% | 0.06% | 2.75% |
| $2^{-4}$ | 10.14% | 2.26% | 0.86% | 1.24% | 1.62% | $1e^{-5}\%$ | $2e^{-5}\%$ | 0.01% |
| $2^{-3}$ | 4.26% | 0.53% | 0.09% | 0.08% | 0.16% | - | - | - |
| $2^{-2}$ | 0.60% | 0.05% | 0.01% | 0.003% | 0.01% | - | - | - |
| $2^{-1}$ | - | $3e^{-4}\%$ | $2e^{-4}\%$ | $3e^{-4}\%$ | - | - | - | - |
| Total | 100% | 100% | 100% | 100% | 100% | 100% | 100% | 100% |
| Bit-width | 4 | 4 | 4 | 4 | 4 | 4 | 4 | 4 |

## B    APPENDIX 2: LOSSLESS CNNS WITH LOW-PRECISION WEIGHTS AND LOW-PRECISION ACTIVATIONS

Table 7: Comparison of our VGG-16 model with 5-bit weights and 4-bit activations, and the pre-trained reference with 32-bit floating-point weights and 32-bit float-point activations.

| Network | Bit-width for weight/activation | Top-1 error | Top-5 error | Decrease in top-1/top-5 error |
|---|---|---|---|---|
| VGG-16 ref | 32/32 | 31.46% | 11.35% | |
| VGG-16 | 5/4 | 29.82% | 10.19% | 1.64%/1.16% |

Recently, we have made some good progress on developing our INQ for lossless CNNs with both low-precision weights and low-precision activations. According to the results summarized in Table 7, it can be seen that our VGG-16 model with 5-bit weights and 4-bit activations shows improved top-5 and top-1 recognition rates in comparison to the pre-trained reference with 32-bit floating-point weights and 32-bit floating-point activations. To the best of our knowledge, this should be the best results reported on VGG-16 architecture so far.

