# Peer review of "Incremental Network Quantization: Towards Lossless CNNs with Low-precision Weights"

_ICLR 2017 — accepted_

[Public Comment · (anonymous) · 26 Nov 2016 (modified: 26 Jan 2017)]
**pre-review questions**

Dear Reviewers,

Please take a look through the paper and ask the authors to clarify any questions you might have. The deadline for this part of the review process is December 2, 2016.

Thanks!

[Official Review · AnonReviewer3 · rating 7 · confidence 4 · 16 Dec 2016]
**Reasonable idea**

The idea of this paper is reasonable - gradually go from original weights to compressed weights by compressing a part of them and fine-tuning the rest. Everything seems fine, results look good, and my questions have been addressed.

To improve the paper:

1) It would be good to incorporate some of the answers into the paper, mainly the results with pruning + this method as that can be compared fairly to Han et al. and outperforms it.

2) It would be good to better explain the encoding method (my question 4) as it is not that clear from the paper (e.g. made me make a mistake in question 5 for the computation of n2). The "5 bits" is misleading as in fact what is used is variable length encoding (which is on average close to 5 bits) where:
- 0 is represented with 1 bit, e.g. 0
- other values are represented with 5 bits, where the first bit is needed to distinguish from 0, and the remaining 4 bits represent the 16 different values for the powers of 2.

[Official Review · AnonReviewer2 · rating 7 · confidence 3 · 16 Dec 2016 (modified: 14 Jan 2017)]
**Quantize a fully trained network with an iterative 3 step process of partition/hard quantize/retrain, repeated on the retrained partition until fully quantized. Achieves nice results on ImageNet tasks down to 4 bits, but is missing pruning steps which is needed for large competitive compression.**

Nice idea but not complete, model size is not reduced by the large factors found in one of your references (Song 2016), where they go to 5 bits, but this is ontop of pruning which gives overall 49X reduction in model size of VGG (without loss of accuracy). You may achieve similar reductions with inclusion of pruning (or better since you go to 4 bits with no loss) but we should see this in the paper, so at the moment it is difficult to compare

[Official Review · AnonReviewer1 · rating 8 · confidence 4 · 19 Dec 2016]
**Great idea, very impressive results.**

There is a great deal of ongoing interest in compressing neural network models. One line of work has focused on using low-precision representations of the model weights, even down to 1 or 2 bits. However, so far these approaches have been accompanied by a significant impact on accuracy. The paper proposes an iterative quantization scheme, in which the network weights are quantized in stages---the largest weights (in absolute value) are quantized and fixed, while unquantized weights can adapt to compensate for any resulting error. The experimental results show this is extremely effective, yielding models with 4 bit or 3 bit weights with essentially no reduction in accuracy. While at 2 bits the accuracy decreases slightly, the results are substantially better than those achieved with other quantization approaches.

Overall this paper is clear, the technique is as far as I am aware novel, the experiments are thorough and the results are very compelling, so I recommend acceptance. The paper could use another second pass for writing style and grammar. Also, the description of the pruning-inspired partitioning strategy could be clarified somewhat... e.g., the chosen splitting ratio of 50% only seems to be referenced in a figure caption and not the main text.

[Author Response · Aojun Zhou · 14 Jan 2017 (modified: 16 Jan 2017)]
**Paper update: all required result comparisons have been added!**

Thanks to all the reviewers for constructive suggestions and comments. We are really excited that the novelty of our paper has been well recognized.

In this updated version, we carefully considered all reviewers’ suggestions to improve the paper. Generally, we performed four aspects of works: (1) the result comparison of pruning + quantization between our method and Han et al.’s method [1] was incorporated into the paper (please see Section 3.4 for details); (2) the result comparison of weight quantization between our method and vector quantization [2] was also incorporated into the paper (please see Section 3.4 for details); (3) we tried our best to improve the clarifications of our encoding method for weight quantization, definition of bit-width, detailed experimental settings and so on, and several rounds of proof-reading and revising were also conducted; (4) more experimental results (including the statistical analyses on the distribution of weights after quantization and our latest progress on developing INQ for deep CNNs with low-precision weights and low-precision activations) that reviewers may be interested were added to the paper as the supplementary materials.
 
Moreover, to make our work fully reproducible, the code (along with an instruction manual) will be released to public as we promised in the paper submission.

(1) To reviewer 1:

Question1: “Also, the description of the pruning-inspired partitioning strategy could be clarified somewhat... e.g., the chosen splitting ratio of 50% only seems to be referenced in a figure caption and not the main text.”

Following your suggestion, we added detailed parameter settings (such as splitting ratio and etc.) to the respective sets of experiments described in Section 3 accordingly.

Question 2: “The paper could use another second pass for writing style and grammar.”

Following your suggestion, we tried our best to do a much better work on revising and proof-reading, with the helps from the native colleagues in USA.

(2) To reviewer 2:

Thanks for your recognition of the novelty of our method. We believe that our responses posted on Dec. 16, 2016 should well address your concern on the result comparison of pruning + quantization between our method and Han et al.’s method [1]. Furthermore, detailed result comparisons can be found in Section 3.4 of the paper. It can be clearly seen that our method outperforms Han et al.’s method [1] with significant margins.

(3) To reviewer 3:

Question 1: “1) It would be good to incorporate some of the answers into the paper, mainly the results with pruning + this method as that can be compared fairly to Han et al. and outperforms it.”

Following your suggestion, we incorporated related results into the paper (please see Section 3.4 for details).

Question 2: “It would be good to better explain the encoding method (my question 4) as it is not that clear from the paper (e.g. made me make a mistake in question 5 for the computation of n2).”

Following your suggestion, we revised related parts, especially the clarification of our encoding method based on our previous responses to your questions accordingly (please see Section 2.1 for details).

Question 3: "The "5 bits" is misleading as in fact what is used is variable length encoding (which is on average close to 5 bits) where: - 0 is represented with 1 bit, e.g. 0 - other values are represented with 5 bits, where the first bit is needed to distinguish from 0, and the remaining 4 bits represent the 16 different values for the powers of 2.”

Following your suggestion, we made a clear clarification on the definition of bit-width accordingly.

References:
Song Han, Jeff Pool, John Tran, and William J. Dally. Deep compression: Compressing deep neural networks with pruning, trained quantization and huffman coding. ICLR, 2016.
Yunchao Gong, Liu Liu, Ming Yang, and Lubomir Bourdev. Compressing deep convolutional networks using vector quantization. arXiv preprint arXiv:1412.6115v1, 2014.

[Final Decision · Program Chairs · 06 Feb 2017]
**ICLR committee final decision**

The paper presents a method for iterative quantization of neural networks weights to powers of 2. The technique is simple, but novel and effective, with thorough evaluation on a variety of ImageNet classification models.